

**Changes in the partial pressure of carbon dioxide in the Mauritanian-Cape Verde**
**upwelling region between 2005 and 2012.**
**By**
**Melchor   González-Dávila[1*], J. Magdalena Santana Casiano[1] and Francisco**
**Machín[1,2]**
**[1]Instituto de Oceanografía y Cambio Global, Grupo QUIMA, Universidad de Las**
**Palmas de Gran Canaria, 35017, Las Palmas de Gran Canaria. Spain.**
**[2]Departamento de Física, Universidad de Las Palmas de Gran Canaria, 35017, Las**
**Palmas de Gran Canaria**
* Correspondence to melchor.gonzalez@ulpgc.es





**ABSTRACT**
Coastal upwelling along the eastern margins of major ocean basins represent regions of
large economic importance due to the high biological productivity. However, the physical
forcing of upwelling processes that favor the production in these areas are being affected
by global warming, which will modify the intensity of the upwelling and, consequently,
the carbon dioxide cycle. For this reason, the role of observations in addressing any
climate change impacts on the global carbon cycle in areas of upwelling is of great
importance. Monthly high resolution surface experimental data for temperature and
partial pressure of carbon dioxide in the Mauritanian-Cape Verde upwelling region from
2005 to 2012 are shown. This data set provides direct evidence of seasonal and
interannual changes in the physical and biochemical processes. They confirmed an
upwelling intensification and an increase in the $CO_2$ outgassing of 1 Tg a year in one of
the four most important upwelling regions of the planet due to wind increase, even when
primary production seems to also be reinforced. This increase in $CO_2$ intake together with
the observed decrease in sea surface temperature at the location of the Mauritanian Cape
Blanc, 21°N, produced a pH decrease of $-0.003 \pm 0.001$ per year.



## 1. INTRODUCTION

The excess of $CO_2$ in the atmosphere, largely responsible of Global Climate Change, has prompted research on the role of the oceans in the carbon cycle. In recent decades data from different oceans have been taken as thoroughly as possible, with the aim to assess how the oceans act as sources or sinks within the carbon cycle. To achieve this goal, observations representative of the distribution of $CO_2$ fluxes between the ocean and atmosphere are necessary. In this regard, automated instruments have been installed on opportunity ships for sampling the ocean as much as possible, data is being collected at scientific cruises and long-term moorings have been deployed in various sites of the oceans (i.e., Astor et al., 2005: Lüger et al., 2004, 2006; González-Dávila et al., 2005; 2009; Schuster et al., 2009; Ullman et al., 2009; Watson et al., 2009; Padín et al., 2010; Gruber et al., 2002; Dore et al., 2003; Santana-Casiano et al., 2007; Bates et al., 2014). With the amount of data already gathered (http://www.socat.info/), climatologies that present average fluxes between the atmosphere and the ocean have been developed, so areas acting as a source or sink are now identified (Key et al., 2004; Takahashi et al., 2009). However, the low spatial resolution of these databases makes it lose relevant variability at relatively low spatial scales. This occurs in coastal areas, specially in upwelling regions, which are not adequately represented in large databases. Upwelling zones present a dynamic that raises water from relatively deep areas, which are rich in nutrients and $CO_2$. Nutrients promote primary production, which consumes $CO_2$, a process that would generate a $CO_2$ flux into the ocean. On the other hand, the upwelling also brings up $CO_2$ from deep seawater, which finally generates uncertainty about the actual role of upwelling areas as a source or sink of $CO_2$ (Michaels *et al.*, 2001). Indeed, previous researches indicate that upwelling areas act as a source or sink of $CO_2$ depending





on their location (Cai *et al.*, 2006; Chen et al, 2013), where upwelling areas at low
latitudes mainly act as a source of $CO_2$ (Feely *et al.*, 2002; Astor *et al.*, 2005; Friederich
*et al.*, 2008; Santana-Casiano *et al.*, 2009; González-Dávila *et al.*, 2009) and those at mid-
latitudes act as a sink of $CO_2$ (Frankignoulle and Borges, 2001; Hales *et al.*, 2005; Borges
*et al.*, 2002; 2005; Santana-Casiano *et al.*, 2009; González-Dávila *et al.*, 2009). Several
anthropogenic interactive effects are strongly influencing the general picture for the most
representative Eastern Boundary Upwelling Systems (EBUS), and include upper ocean
warming, ocean acidification and ocean deoxygenation (Gruber, 2011; Feely et al., 2008;
Keeling et al., 2010). Moreover, evidence for an increase in winds that favor upwelling
(Bakun, 1990; Demarcq, 2009; Oerder et al. 2015) support the possibility of a change in
the current role of these highly productive areas. Recently, eddy-resolving regional ocean
models have shown how upwelling intensification can be followed by major impact on
the system's biological productivity and in the $CO_2$ outgassing (Lachkar and Gruber,
2013; Oerder et al., 2015). Wind observations and reanalysis products are controversial
regarding the Bakun intensification hypothesis (Bakun 1990). Using different winds
database for the Canary region, Barton et al. (2013) concluded that there was no evidence
for a general increase in the upwelling intensity off northwest Africa. Marcello et al.
(2011) found an intensification of the upwelling system in the same area during a 20-year
period while the alongshore wind stress remained almost stable. Cropper et al. (2014)
found that coastal summer wind speed increased, resulting in an increase in upwelling-
favorable wind speeds north of 20ºN and an increase in downwelling-favorable winds
south of 20ºN. Santos et al (2005; 2012) showed differences in Sea Surface Temperature,
SST, between coast and ocean depending on the upwelling index, UI intensity, and that
SST trends were not homogeneous either along latitude or longitude. Varela et al. (2015)
also showed opposite results world wide when different wind databases were used and



when the same wind database was considered depending on the length of data, season
evaluated, and selected area. For the Mauritanian region, when wind stress data were used
(Varela et al., 2015), a more persistent increasing trend in upwelling-favourable winds
north of 21ºN and a decreasing trend south of 19ºN were determined.
Starting in June 2005, the QUIMA-voluntary opportunity ship line visited the
Mauritanian-Cape Verde upwelling region northwest of Africa on a monthly basis (Fig.
1 and Supplementary Table 1S) producing for the first time a high resolution SST and
partial pressure of $CO_2$, $fCO_2$, database. This database has been considered to show the
variations in the $CO_2$ system under changes in the upwelling conditions in the Canary
Ecosystem from 27ºN to 10ºN for the period 2005 to 2012.

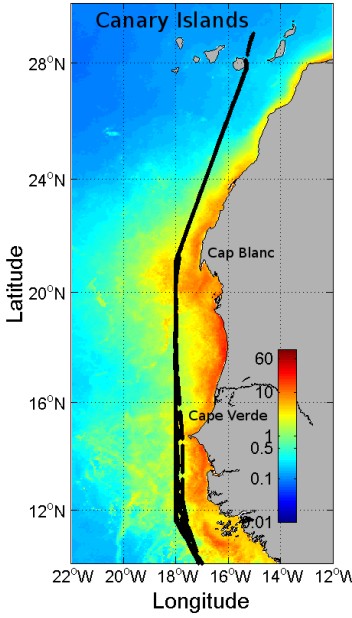

Fig. 1. Ship track in the area from 28ºN (Gran Canaria, The Canary Islands) to 10ºN (black dots).
The locations of Cap Blanc and Cape Verde are indicated. Monthly Ocean Color
(oceancolor.gsfc.nasa.gov) data for average chlorophyl $a$ concentration (mg m$^{-3}$) were included
in a MatlabTM routine and annually averaged, in order to draw the map for the area. The map has
been generated using Matlab 7.12 R2011a.





## 2. EXPERIMENTAL

**2.1    Region evaluated.**

The VOS line crosses the East Atlantic Ocean from the north of Europe (English Channel) to South Africa, calling Gran Canaria, the Canary Islands, with a periodicity of two months, which provides monthly data (southward or northward sections). In this work, the area between Gran Canaria at 27ºN and 10 ºN has been selected in order to study the Mauritanean-Cape Verde upwelling region. In its south route (Fig. 1), the ship leaves Gran Canaria, and goes straight to 100 Km off Cap Blanc, at 21ºN 17º45'W. It then follows this longitude, passing at 100 km off Cape Verde until 12ºN, where it changes direction to Cape Town, reaching 10ºN 17ºW at 330 km out of the coast of Guinea. Between 22ºN and 20ºN, the ship reaches the 500 m isobath. South of 15ºN, the ship moves between 1000 and 500 m isobath. In its north route, the ship follow the same reverse track.

**2.2    Experimental data**

Experimental data were obtained under the EU projects Carboocean and Carbochange (www.CarboOcean.org, https://carbochange.b.uib.no/) and now also available at http://www.socat.info/. An autonomous instrument for the determination of the partial pressure of $CO_2$ developed by Craig Neill following NOAA recommendations was installed in opportunity ships operated by the MSC company during the 2005 to 2008 period and Maersk Company from 2010 to 2012 along the so called QUIMA-VOS line between the UK and Cape Town, from July 2005 to January 2013 (Supplementary Table S1). Temperature was measured at three locations along the sampling circuit: in the intake (SeaBird SBE38L), in the equilibrator (SeaBird thermosalinograph SBE21 and internal PT100 thermometer), and in the oxygen sensor (Optode 3835 Aanderaa$^{TM}$). Differences between equilibrator and intake were constant in time due to the high



seawater flux used but varied among ships due to the different locations of the equipment.
Values varied between 0.06ºC when the equipment was placed close to the intake to
0.35ºC, when the equipment was one floor above, inside the engine room. The SST was
also obtained from the NOAA_OI_SST-V2 data provided by the NOAA/OAR/ESRL
PSD, Boulder, Colorado, USA (http://www.esrl.noaa.gov/psd). These data had a spatial
resolution of 1º latitude and 1º longitude and monthly averages were used. The correlation
between our experimental SST data and satellite one was better than $\pm$ 1ºC, and reduced
to $\pm$ 0.4ºC after removing the most affected upwelling regions (19-22ºN and 14-16ºN) ,
related to the high variability impossed by the upwelling.
The $CO_2$ molar fraction, $xCO_2$, in seawater was obtained every 150 s, while
atmospheric $xCO_2$ data were taken every 200 min. The seawater intake was located at a
10 m depth. The system was calibrated every three hours, by measuring four different
standard gases with mixing ratios of 0.0, 250 ppm, 380 ppm and 490 ppm of $CO_2$ in the
air, provided by NOAA and traceable to the World Meteorological Organisation scale.
The precision of the system is greater than 0.5 µatm and the accuracy estimated with
respect to the standard gases is of 1 µatm. The fugacity of $CO_2$, $fCO_2$ (µatm), was
calculated from $xCO_2$ after correcting for temperature differences between intake and
equilibrator, according to the expressions for the seawater given by DOE (1994).
In order to compute a second carbonate system variable, the surface total alkalinity
was computed from sea surface salinity, SSS, and SST (Lee et al., 2006). $pH_T$ at the in
situ temperature was computed from $fCO_2$ and $A_T$ and with average annual surface ocean
total phosphate and total silicate concentrations of 0.5 and 4.8 µmol kg$^{-1}$, respectively,
from the World Ocean Atlas 2009, using the carbonic acid acidity constants by Merbach
et al (1973) refitted by Dickson and Millero (1987).
Air-sea $CO_2$ fluxes, $FCO_2$ (mmol m$^{-2}$ d$^{-1}$), were evaluated as





$$FCO_2 = 0.24 * k * s * (fCO_2^{sw} - fCO_2^{atm})$$ (1)
where 0.24 is the scale factor, $k$ is the gas transfer velocity, $s$ is the $CO_2$ solubility, $fCO_2^{sw}$
is the seawater fugacity of $CO_2$ and $fCO_2^{atm}$ is the atmospheric fugacity of $CO_2$. In order
to evaluate $\Delta fCO_2$ ($\Delta fCO_2 = fCO_2^{sw} - fCO_2^{atm}$), $fCO_2^{atm}$ data were linearly interpolated to
the $fCO_2^{sw}$ time vector. A positive value for $FCO_2$ corresponds with a $CO_2$ outgassing
from the ocean. $k$ (cm h$^{-1}$) was evaluated with the parametrization (Wannikhoff, 1992):
$$k = 0.31 * W^2 * (Sc/660)^{-1/2}$$ (2)
where $W$ is the wind speed at 10 m above the sea surface (m s$^{-1}$) and $Sc$ is the Schmidt
number.
The variables involved in estimating $FCO_2$ data (i.e. $fCO_2^{sw}$, $fCO_2^{atm}$, SST and SSS)
were fitted to sinusoidal expressions (Lüger et al.,2004) for a given latitude as:
$$X(lat)^* = a_0 + a_1(t - 2005) + a_2 sin(2\pi t) + a_3 cos(2\pi t) + a_4 sin(4\pi t) +$$
$a_5 cos(4\pi t)$ (3)
where $a_i$ are the fitting coefficients, $t$ is the sampling time expressed as year fraction and
$X$ represents any of the four variables. This procedure allowed us to re-construct the series
of experimental data for periods not properly sampled. The variables were decomposed
into an interannual term $X(lat)_t^* = a_0 + a_1(t - 2005)$ plus a periodical term $X(lat)_p^* =$
$a_2 sin(2\pi t) + a_3 cos(2\pi t) + a_4 sin(4\pi t) + a_5 cos(4\pi t)$, that is, $X(lat)^* = X(lat)_t^* +$
$X(lat)_p^*$. The periodical term accounts for the high frequency seasonal variability, while
the interannual one marks the year-to-year trend. First, observations were grouped in a
natural year for a given latitude, as if they had been taken in a single year (no correction
was done for interannual variability). The mean seasonal climatology data associated with
the periodic coefficients (i.e. $a_2$, $a_3$, $a_4$, and $a_5$) throughout the sampling period were
determined. Next, the interannual coefficients $a_1$ were calculated by fitting the residuals
resulting from subtracting the periodical component, $X(lat)_p^*$, from the original variable





$X(lat)$. Fixing these five coefficients ($a_1$-$a_5$), new distributions for $fCO_2^{sw*}$, $fCO_2^{atm*}$, $temp^*$
and $salinity^*$ were constructed with a daily resolution based on the curve fits given for
each variable as in Eq. (3), providing the coefficient $a_0$. The mean residual on the
determination of those four variables with respect to the experimental values were $\pm$ 3.7
μatm, $\pm$ 1.5 μatm, $\pm$ 0.22 ºC, and $\pm$ 0.05 for $fCO_2^{sw*}$, $fCO_2^{atm*}$, $temp^*$ and $salinity^*$,
respectively. When the monthly satellite SST values were considered, the new $temp^*$
function averaged for each month produced values within $\pm$ 0.47ºC, confirming that this
procedure was able to fit non-sampled periods. It was assumed that the same procedure
was valid for non-sampled $fCO_2$. Finally, daily $FCO_2^*$ time series between 10 and 27ºN
with a latitudinal resolution of 0.5° were calculated with a standard error of estimation of
0.5 mmol m$^{-2}$ d$^{-1}$ (15% of error) that produced mean residuals (experimental $FCO_2$ -
$FCO_2^*$) of 0.4 mmol m$^{-2}$ d$^{-1}$ and Pearson correlation coefficients between experimental
and computed $FCO_2^*$ of r > 0.6, p < 0.01.

Chlorophyll-a was calculated from measurements made by the Moderate Resolution

Imaging Spectroradiometer (MODIS) aboard NASA's Aqua satellite. Monthly averages
with spatial resolution of 9 km supplied by Ocean Color (oceancolor.gsfc.nasa.gov) were
used.

Wind    data    were    downloaded    from    the    NCEP    CFSR    database    at

http://rda.ucar.edu/pub/cfsr.html developed by NOAA and retrieved from the NOAA
National Operational Model Archive and Distribution System and maintained by the
NOAA National Climatic Data Center. The spatial resolution is approximately 0.3 × 0.3°
and the temporal resolution is 6 hours. The reference height of the wind data is 10 m.

Rainfall data were collected by the Precipitation Radar installed on the Tropical

Rainfall Measuring Mission (TRMM) satellite (http://precip.gsfc.nasa.gov). Monthly
averages with a spatial resolution of 0.5º×0.5º (product 3A12, version 07) were used




(Supplementary Fig. S1) in order to explain changes in seasonal surface salinity
distributions.

**3.   RESULTS AND DISCUSSION**
**3.1    Physical propeties**
The variability of the Mauritanian-Cape Verde upwelling was analyzed in terms
of the upwelling index (Nykjaer and Van Camp, 1994) (Fig. 2) using satellite wind data.

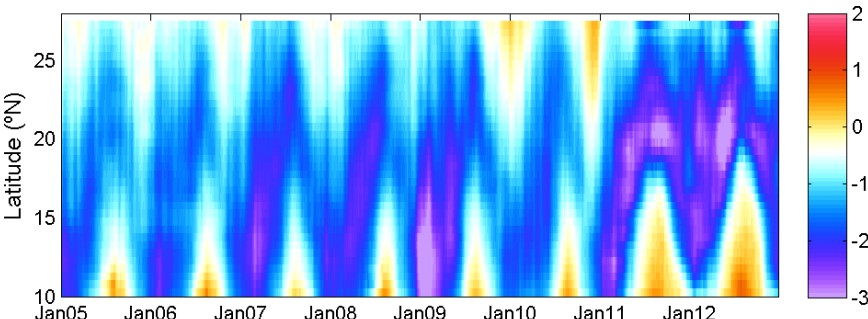

Fig. 2. Time series of upwelling index ($m^2$ $s^{-1}$) in the Mauritanian-Cape Verde upwelling region
along the ship track computed following Nykjaer and Van Camp (1994).

Results clearly distinguish two main subareas in the upwelling system. 1) North
of 20ºN, the upwelling conditions were favorable throughout the year, although the
highest upwellings were observed from March to September with a northward shift from
20º to 22ºN. 2) South of 20ºN, a marked seasonality was observed. South of 15ºN, in the
Cape Verde area, upwelling conditions were favorable during autumn and winter with the
maximum intensity observed during January and February due to the replacement of the
trade winds during the summer by the monsoon winds, which advect warm water
northward along the shore (Nykjaer and Van Camp, 1994). Our results (Fig. 2) are quite
consistent with previous research (Nykjaer and Van Camp, 1994; Marcello et al., 2011;
Santos et al., 2005; 2012) but include the years 2010 to 2012 where UI at around 20-21ºN





presented a shift of the upwelling regime intensity from high to strong. The analysis of
upwelling trends along this area has been controversial since it is highly dependent on the
selected region (Santos et al., 2012). The inter-annual evolution of UI over the period
2005 to 2012 (Fig. 3, green line) determined by averaging monthly values on an annual
base followed that showed by Santos et al. (2012), indicating an increase in the UI (mean
confidence interval of 9 $m^2\,s^{-1}$).

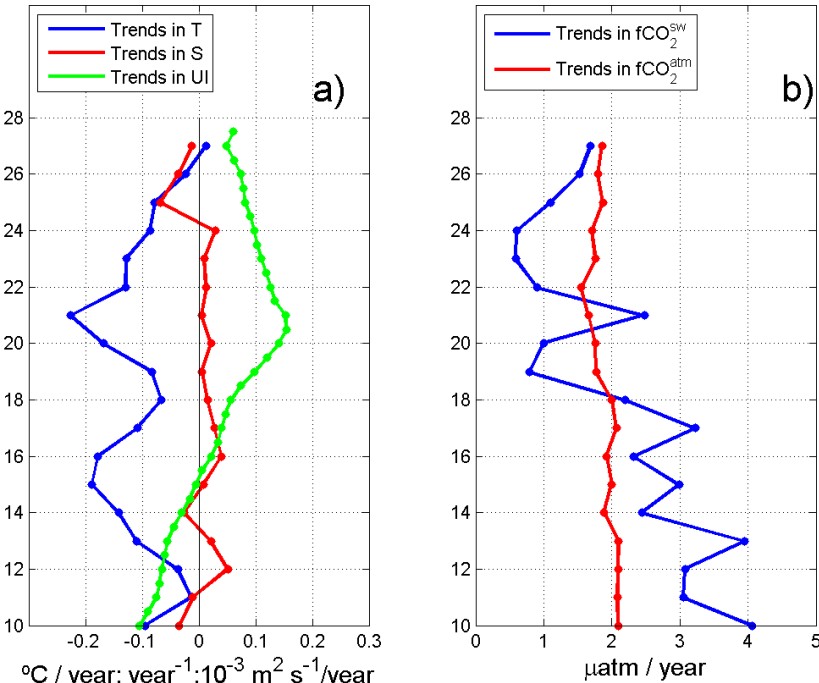

Fig. 3. Latitudinal distribution of the interannual trends for the Upwelling Index (UI) and for the
four experimental variables along the QUIMA-VOS line integrated over every degree. The a)
panel presents the trends for Upwelling index (mean confidence interval of 9 $m^2s^{-1}$), SST (ºC
$yr^{-1}$, confidence interval 0.13ºC) and SSS ($yr^{-1}$, confidence interval 0.06) and the b) pannel the
trends for $fCO_2{}^{sw}$ and $fCO_2{}^{atm}$ (confidence intervals 4.23 and 0.44 µatm).


The upwelling index (except for the area south of 15ºN) confirmed the stronger

upwelling observed since 1995-1996 in this region after a more than a 10-year (from at
least 1982 to 1995) period of weaker upwelling (Santos et al., 2012). Local zonal



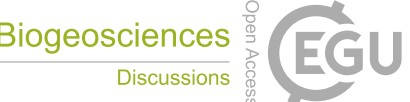

differences between ocean and coastal SST trends determined by using satellite data by
Santos et al. (2005) for the period 1982 to 2000, extended by Santos et al. (2012) until
2010 and in this study until 2012 (data not shown) confirmed the intensification of the
upwelling regime along the African coast. This has been described as a decadal scale shift
of the upwelling regime intensity (Marcello et al., 2011; Santos et al., 2012). To the south
of 15ºN, the annual UI values (Fig. 2 and 3) indicate that the SST close to the coast is
becoming warmer. They serve as an indication of decadal variability of the summer
monsoon winds and associated northward advection of warm water along the coast
(Santos et al., 2012). The highest upwelling intensity along the VOS line was located at
the capes, Cap Blanc and Cape Verde. From satellite chlorophyll-*a* data, especially off
Cap Blanc, giant filaments with chlorophyll concentrations above 1 mg m$^{-3}$ persist year-
round, spreading from the coast several hundred kilometers offshore (Fig. 1). North of
Cap Blanc the upwelled water originates from the North Atlantic Central Water, and
mixes with South Atlantic Central Water, SACW, towards the south (Mittelstaedt, 1983).
South of Cap Blanc, the upwelling of nutrient rich SACW promotes phytoplankton
growth between Cap Blanc and Cape Verde. Towards 12°N, upwelling is also fed by the
North Equatorial Under Current (Hagen and Schemainda, 1984). Moreover, the entire
northwest African coast is also influenced by the African desert dust transport by the mid-
tropospheric Harmattan winds originating from the central Sahara, which supplements
the levels of micronutrients (such as iron) to the adjacent marine ecosystem (Mittelstaedt,
1983; Neuer et al., 2004; Swap et al., 1996).

The area is also affected by the migration of the Inter-Tropical Convergence Zone

(ITCZ), related to maximum precipitation rates. To have a significant satellite
precipitation record in our region of interest, precipitation data were integrated
longitudinally between 25.25ºW and 9.75ºW. Time series for the latitudinal distribution





of integrated precipitation (Supplementary Fig. 1S) identified the average position of the
ITCZ related to maximum precipitation rates. The ITCZ was located at its southernmost
position (2ºN) during winter, reaching its northernmost position (14-16ºN) around
summer. The ITCZ reached our area of interest (>10°N) from late spring to late summer.

The latitudinal distributions of experimental surface temperature and salinity

along the vessel track are shown in Fig. 4, grouped by seasons. In situ temperature at
27°N shows temperatures in the range of 18 to 24°C with the minimum in winter and
maximum in late summer-early autumn. The annual temperature range was somewhat
higher at 20°N, with summer maximum of around 26°C and minimum in spring of about
17ºC. At 10°N, temperatures were the highest throughout the year (>25°C), with
minimum values in winter and maximum in late spring and late autumn. The low values
observed during the end of summer are related to the arriving of the ITZC at those
latitudes. The thermal distribution shows a temperature increase as we move to the
Equator and a notable cooling at the upwelled waters off Mauritania. The temperature
generally decreased from 10ºN to about 20ºN to 21ºN, where the ship meets the
Mauritanian upwelling. From there to the north, the temperature rises as the ship leaves
the upwelling area on its way to the Canary Islands. Only during winter time and the
begining of the spring, the upwelling of cold water from Cape Verde area was detected.
Salinity minimum values were normally located at 10ºN, increasing to maximum values
at the Canaries' latitude. The minimum values of salinity were exceptionally low during
autumn from 10ºN to 16ºN by both the freshwater input from rivers that increase their
outflow during this season (Nicholson, 1981) and by the northward shift of the ITCZ
during this part of the year.






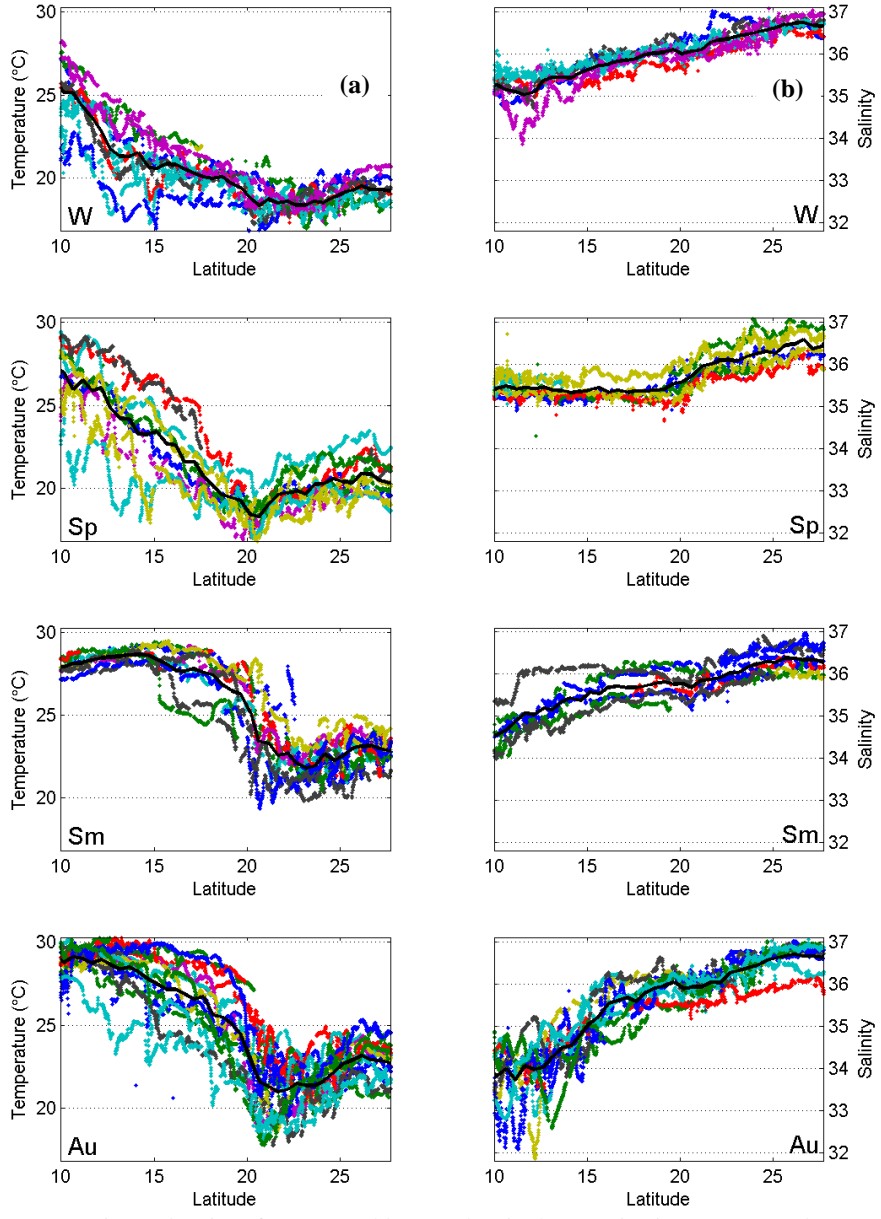

Fig. 4. Experimental series of a) SST and b) SSS data in the Mauritanian - Cape Verde coastal region grouped by seasons: winter (W), spring (Sp), summer (Sm) and autumn (Au). The averaged values for all cruises in Table S1, are shown in black for each season including the 95% confidence limits.

Anomaly fields for temperature and salinity (data not shown) were calculated as

the difference between the observations and the mean values at each season for individual





latitudes. For temperature, the largest anomalies in winter and spring were located south
of 18°N, with values of ±2ºC, related to the seasonal cycle of the Cape Verde upwelling.
During summer the pattern changed and the largest anomalies were detected in the
upwelling area at 18-22°N, with values of ±5ºC when the upwelling index for the
Mauritanian area was highest (Fig. 2). In autumn the temperature anomalies were shifted
slightly to the north, 20-24°N, with values of ±3ºC related to the observed pulses in
upwelling favorable winds that affected the surface seawater properties. On the other
hand, salinity anomalies showed a very homogeneous pattern in all latitudes for winter,
spring and summer, with values generally within ±0.5. However, during autumn
important anomalies south of 18°N were observed, with values in the range of ±1.5. In
this region, the upwelling development, the river discharge and the rainy season
controlled the observed distribution (Yoo and Carton, 1990).

**3.2    Carbon dioxide variability**
The latitudinal distribution of the experimental $fCO_2^{sw}$ data (Fig. 5a) grouped by
seasons, showed they were always above the $fCO_2^{atm}$, with the highest values between
18 and 23ºN for all seasons due to the variability imposed by the upwelling off
Mauritania. During winter, when the Cape Verde upwelling develops (Fig. 2), the 12-
15ºN region also presented higher $fCO_2^{sw}$ values than those in the atmosphere. $fCO_2^{sw}$
data showed a latitudinal shift along the sea1sons following that observed in the upwelling
index: i.e., in winter, the largest values were located between 19º and 24ºN; in spring,
they were located between 16º and 22ºN; during summer and autumn, the largest $fCO_2^{sw}$
values were recorded in the range 20º to 23ºN. $fCO_2^{sw}$ normalized to the mean SST of
22ºC for the region (N$fCO_2^{sw}$, Fig. 5b) reinforced the variability indicating that upwelling
is the major factor contributing to the $fCO_2$ variability.





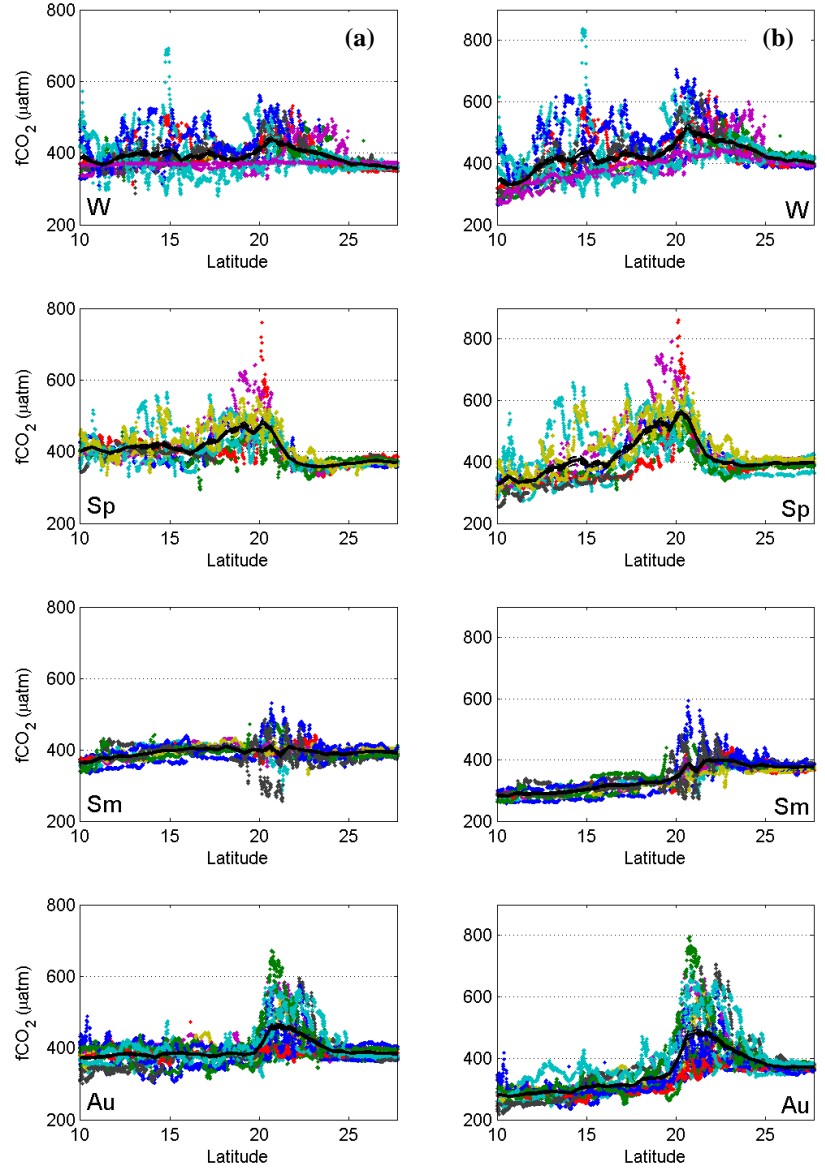

Fig 5. Experimental series of fugacity of $CO_2$ data in the Mauritanian-Cape Verde coastal region grouped by seasons: winter (W), spring (Sp), summer (Sm) and autumn (Au). a) $fCO_2^{sw}$ latitudinal distribution. b) Normalized $fCO_2^{sw}$ values to a constant temperature of 22ºC. The averaged values for all cruises in Table S1, are shown in black for each season including the 95% confidence limits.

According to Takahashi et al. (1993), $fCO_2^{sw}$ increases with temperature at a rate that is 4.3% µatm ºC$^{-1}$ (around 16 µatm ºC$^{-1}$ at this area) in a thermodinamically controlled

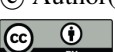



system. At 27ºN, the rate, as SST increases, was only of 7.45 µatm ºC$^{-1}$ due first to
biological uptake and second to the $CO_2$ outflux. At 20ºN the rate became negative with
a value of -10.9 µatm ºC$^{-1}$, clearly indicating the important injection of cool and $CO_2$ rich
seawater at the upwelling area, neither being compensated by the solubility nor the
biological carbon pumps. At 10ºN, and as a result of the seasonal upwelling, the rate was
still negative but of only -4.3 µatm ºC$^{-1}$. When $NfCO_2^{sw}$ was related with SST at the
latitudes 19º to 21ºN in the upwelling vicinity of Cap Blanc, an inverse relationship of
70-100 µatm ºC$^{-1}$ was found during winter and spring, while in summer and autumn the
inverse relationship was reduced to 12-18 µatm ºC$^{-1}$. While the upwelling indexes at those
latitudes were quite constant throughout the year, the different rates observed should be
related to biological consumption of the $CO_2$ excess. During winter and spring the
injection of $CO_2$ in the upwelling is not decreased by the biological activity in the area.
At the end of spring and during summer the Chl-*a* content reached its maximum and most
of the $CO_2$ was consumed and/or exported and, therefore, the rate was strongly reduced.

Figure 3 depicts the observed interannual trends ($a_1$ coefficient in Eq. 3) for the

four experimental recorded detrended parameters, together with the UI trend. Confidence
intervals of the computed mean annual values for SST, SSS, $fCO_2^{atm}$, $fCO_2^{sw}$ were 0.13ºC,
0.06, 0.44 µatm and 4.23 µatm, respectively. There was a clear SST trend whereby
seawater along the VOS line track was getting cooler with maximum cooling rates at the
location of Cap Blanc (21ºN) and Cape Verde upwellings (15ºN) with rates higher than -
0.2ºC yr$^{-1}$. Data from the first three years (2005 to 2008) at 21ºN showed lower
temperatures with higher cooling rates that reached -0.7ºC yr$^{-1}$, although three years of
data are not representative. The area crossed by the VOS line along 17º45'W from 22ºN
to 10ºN is located inside the 1000 m isobath that is well inside the mean frontal activity
in the Canary region of about 200 km width (Wang et al., 2015). The different changes



in temperature in the coastal slope and offshore waters are related to the different origins
of the waters upwelled from depths of about 100 m to the surface (Mittelstaedt, 1983)
that spread off the coastal area. The offshore water SST is less variable owing to longer
residence time in the ocean surface. These effects and the fact that the VOS line keeps a
track line that crossed the upwelling cells at a distance to the coast that varies among cells,
contributes to the observed spatial variability. There was no attempt to compare
latitudinal and longitudinal effects on the observed values. Our experimental data,
however, does not show any positive annual based SST rates in the upwell affected area,
and only when the ship approached the Canary Islands, the trends became less negative,
reaching a value of +0.02ºC yr$^{-1}$ at 27ºN, similar to those obtained for oceanic Atlantic
water (Bates et al., 2014).
$fCO_2^{atm}$ for the area presented the interannual increase of about $2 \pm 0.3$ µatm y$^{-1}$
observed in atmospheric stations, while $fCO_2^{sw}$ presented a heterogeneous distribution.
South of 18ºN the rate of increase was always higher than that in the atmosphere reaching
a maximum value of $4.1 \pm 0.4$ µatm y$^{-1}$ at 10ºN. At 27ºN, $fCO_2^{sw}$ increased at a rate of
$1.7 \pm 0.2$ µatm y$^{-1}$ similar to that determined at the ESTOC time series site (González-
Dávila et al., 2010) located at 29º10' N 15º30'W. In the Cap Blanc area, $fCO_2^{sw}$ increased
at an averate rate of $2.5 \pm 0.4$ µatm y$^{-1}$ with the highest values in the period 2005 to 2008
(a rate of $4.6 \pm 0.5$ µatm y$^{-1}$ was computed with only those years). Around Cap Blanc,
$fCO_2^{sw}$ always presented lower rates of increase than in the atmosphere with values well
below 1 µatm y$^{-1}$. The observed decrease in SST and the trends in $fCO_2^{sw}$ can only be
explained by a reinforced upwelling. North of 18ºN, the lowest rate of increase in $fCO_2^{sw}$
compared to $fCO_2^{atm}$, together with a decrease in temperature, indicated that upwelling is
also favoring an increase in the net community production around the Mauritanian
upwelling, consuming and/or exporting the $CO_2$ rich upwelled waters favored by the





lateral transport of the Mauritanian current (Lachkar and Gruber, 2013; Varela et al.,
2015). The upwelling intensification effects observed in the trends of our experimental
data support the recent wind stress trends (Crooper et al., 2014; Varela et al., 2015; Santos
et al., 2012) of increased upwelling-favorable winds, at least for the period 2005-2012 in
the Canary upwelling region (Fig. 2 and 3). The intensification of the upwelling results
in a change in the measured upwelled water properties due to either higher upwelling
velocities or deeper source upwelled waters. However, what remains unclear from these
records is to what extent those changes reflect upwelling variations due to climate change
forcing versus natural decadal variability in the upwelling areas occurring over
interannual timescales.

Because of the upwelling intensity is changing, other variables will also be

affected. $pH_{T,is}$ at $21 \pm 0.25$ºN was computed from $fCO_2$ and alkalinity pairs of data.
Alkalinity was computed from regional correlations with SST and SSS (Lee et al., 2006)
which could under-represent seasonal and interannual variations in upwelling areas.
However, pH computed from $fCO_2$ values are relatively insensitive to errors in $A_T$, and
$fCO_2$ controls the magnitude and variability of pH (a 60 µmol kg$^{-1}$ change in $A_T$ will affect
a 0.1% in pH, that is, about 0.01 pH units). Figure 6 depitcs the computed $pH_{T,is}(A_T, fCO_2)$
data and the harmonic fitting Eq. (3) providing seasonal variability and interannual trend.
Considering the small systematic biases in interannual dynamics, it is determined a
decrease in pH at a rate of $-0.003 \pm 0.001$ per year (Fig. 6). This decrease is on the highest
rate values determined in several time series stations (Bates et al., 2014), where oceanic
SST has only slightly increased in the last decades. However, at the Mauritanian
upwelling area and at the location where our VOS line approached this region, SST
decreased at a rate of $-0.22 \pm 0.06$ºC y$^{-1}$ (Fig. 3). Solely, this decrease in temperature
would increase the pH by a rate of $+0.004$ yr$^{-1}$ and the $fCO_2$ would decrease by 4 µatm yr$^{-}$



[1] The net effect of the increase in the amount of rich $CO_2$/low pH upwelled waters in the
Mauritanian upwelling would be, therefore, a decrease in the pH of over $-0.007\pm0.002$
units $y^{-1}$ and an increase in $fCO_2$ of $+6.5 \pm 0.7$ µatm $y^{-1}$ (with periods where those rates
could reach values of 0.015 $y^{-1}$ in pH and 10.5 µatm $y^{-1}$ in $fCO_2$ as recorded during 2005-
2008). Those values are greatly compensated by the important decrease in the SST
resulting in the observed rates of $-0.003 \pm 0.001$ pH units and $+2.5 \pm 0.4$ µatm of $fCO_2$
per year.

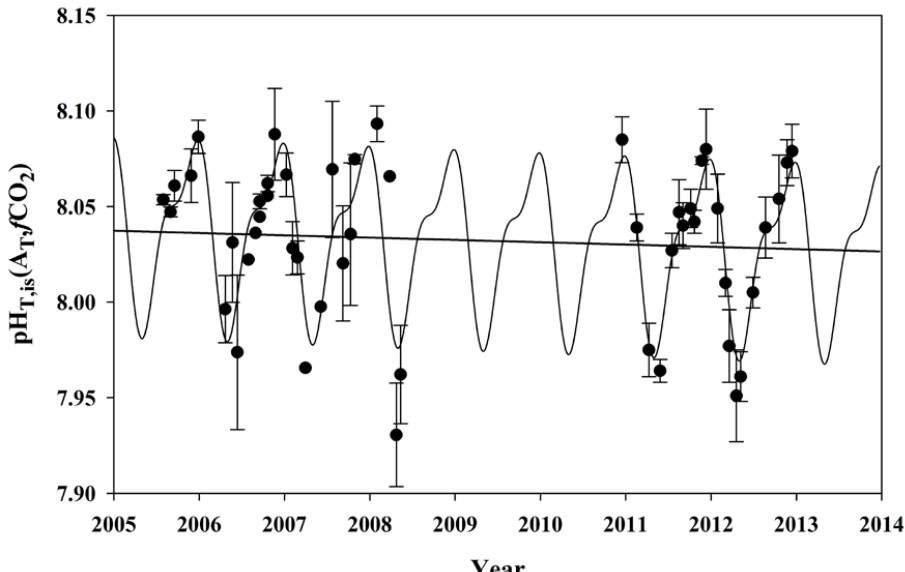

Fig. 6. pH at *in situ* SST in total proton scale computed from total alkalinity (based on regional
correlations with SST and SSS, Lee et al., 2006) and $fCO_2$ at $21 \pm 0.25$ ºN. The error bar represents
the standard deviation of the computed data for each cruise for the selected latitude. The black
line shows the harmonic fitting Eq. (3) for the data and the corresponding linear trend.


## 3.3    Fluxes of $CO_2$
The annual air-sea $CO_2$ flux for the full domain was positive (Fig. 7a), with the area off
Mauritania, between 18 and 22°N, acting as an active source of $CO_2$ to the atmosphere

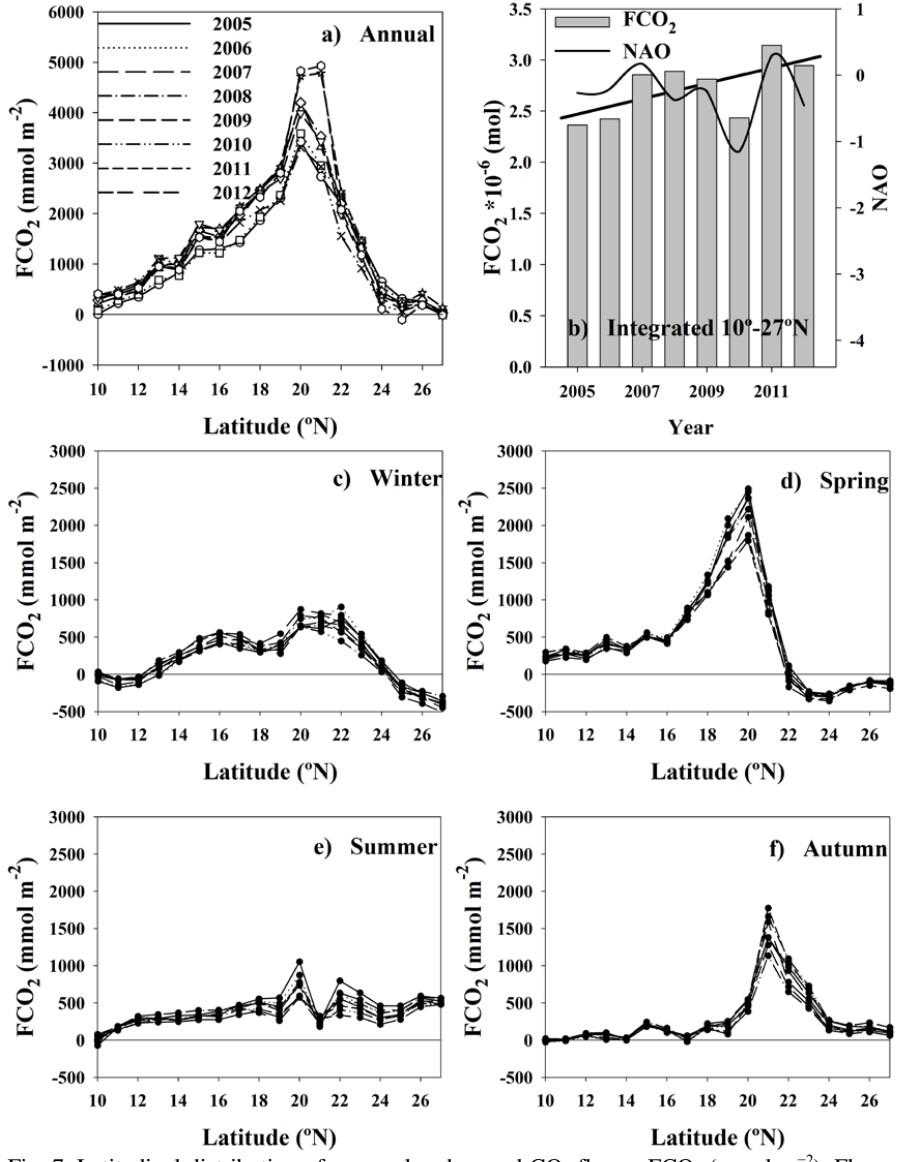

Fig. 7. Latitudinal distribution of seasonal and annual $CO_2$ fluxes, $FCO_2$ (mmol m$^{-2}$). Fluxes of $CO_2$ were computed using Wannikhoff (1992) parametrization and satellite winds with a resolution of 6 hours. a) Integrated year-to-year from 2005 to 2012 and b) latitudinally integrated for 2005 to 2012 together with annual values for NAO index. Latitudinal distribution of $FCO_2$ seasonally integrated from 2005 to 2012 are depicted for winter (c), spring (d), summer (e) and autumn (f).

with values close to 5 mol $CO_2$ m$^{-2}$ (Fig. 7a). North of 24ºN, in the area not affected by

the coastal upwelling, an average flux of +0.2 ± 0.1 mol $CO_2$ m$^{-2}$ was determined.

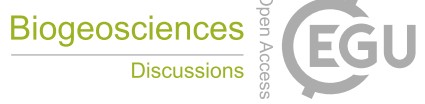

The ingassing observed during winter and spring of -185 ± 36 mmol $CO_2$ m$^{-2}$ for the full
period (Fig. 7) was surpassed by the outgassing during summer and autumn of 295 ± 150
mmol $CO_2$ m$^{-2}$. South of 24ºN, it was observed that during spring (Fig. 7d) the
photosynthetic activity was not intense enough to uptake the $CO_2$ injected by the strongest
upwelling in the surface waters and the area acted as a source of $CO_2$ with values reaching
3 mol $CO_2$ m$^{-2}$ in 2012. During summer (Fig. 7e), primary producers and lateral advection
of warm waters by the Mauritanian current could consume/export the $CO_2$ rich waters
reaching values of 0.5 mol m$^{-2}$. During autumn (Fig. 7f), only the area between 20ºN and
23ºN acted as a source of 1-2 mol $CO_2$ m$^{-2}$, while the rest was almost in equilibrium. It
was also detected that the late autumn-winter upwelling in the 14º to 17ºN region
contributed to an increased outgassing with a second annual submaximum of about 0.7
mol $CO_2$ m$^{-2}$ in winter (Fig. 7c). South of 14ºN, annual $CO_2$ fluxes decreased from about
1 mol m$^{-2}$ at 14ºN to being roughly in equilibrium at 10°N.
The integrated $CO_2$ fluxes for the area 10º to 27ºN along the VOS line section for
the years 2005 to 2012 (Fig. 7b) were between 2.3 and 3.1 10$^6$ mol, with an important
interannual variability. FCO$_2$ increased during the studied period by 0.08 ± 0.03·10$^6$ mol
yr$^{-1}$. The augment in FCO$_2$ is related to the observed increase in wind speed (Fig. 3,
indicated as UI) north of 16ºN that far surpassed the effect of the smaller annual rate of
increase in $f$CO$_2$$^{sw}$ than in $f$CO$_2$$^{atm}$ north of 19ºN with the exception at 21ºN (Fig. 3). South
of 16ºN, the decrease in wind speed did not exceed the effect of the increment in Δ$f$CO$_2$
associated to the increase in downwelling indexes (Fig. 3; Santos et al., 2012), and FCO$_2$
was slightly ascending. The variability observed in the annual integrated $CO_2$ fluxes (Fig.
7b) was related with the basin-scale oscillations, the North Atlantic Oscillation (NAO)
index and the East-Atlantic Pattern (EA)
(http://www.cpc.ncep.noaa.gov/data/teledoc/telecontents.shtml). Cropper et al. (2014)



found winter upwelling variability was strongly correlated with the winter NAO (r values
ranged from 0.50 at 12–19ºN to 0.59 at 21–26ºN), due to the strength of the Azores semi-
permanent high-pressure system, which modifies trade wind strengths. The annual
integrated $FCO_2$ was related with the annual NAO index (Fig. 7b) with a similar r = 0.54,
even when fluxes are not only controlled by wind strength. However, Fig. 7a clearly
indicates that the Mauritanian upwelling area was the most important contributor to
$FCO_2$. There was not a significant correlation coefficient with the winter NAO (r = 0.23).
Also, the EA index, because represents the southward-shifted NAO-like oscillation,
presented a lower significant value (r = 0.48), as it was observed by the upwelling index
(Cropper et al., 2014). The correlation between fluxes and climate indexes describing the
main mode of variability across the Atlantic sector may be directly related to the Azores
High and its influence on the trade wind strength.
If the $FCO_2$ values are assumed to be valid for at least 100 km to both sides of the
QUIMA-VOS line, the total flux of $CO_2$ being ejected to the atmosphere would reach a
value of 30 Tg of carbon dioxide a year for the period 2005-2012, with a rate of increase
of 1 Tg $yr^{-1}$. However, it should be considered that the export of the rich $fCO_2$ upwelled
water with high nutrient concentration off the coastal areas would promote a decrease in
surface $fCO_2$ values (as those observed north and south 21ºN) that will produce an
ingassing of $CO_2$. This could balance the observed outgassing increase in a more global
scale.

## 4. CONCLUSIONS

The Mauritanian-Cape Verde upwelling has been shown to be an important area sensitive
to decadal and climate change forcing on upwelling processes, which strongly affects the
$CO_2$ surface distribution, ocean acidification rates and air-sea $CO_2$ exchange.



The results for the period 2005 to 2012 confirm, firstly, the upwelling intensification at
the Mauritanian-Cape Verde upwelling system by using experimental SST and carbon
dioxide system variables. Secondly, that upwelling regions at low-mid latitudes are strong
sources of $CO_2$ to the atmosphere and thirdly, that as a direct result, the pH is decreasing
at a rate of $-0.003 \pm 0.001$ per year and the amount of emitted $CO_2$ is increasing annualy
at a rate of 1 Tg due to wind increase even when primary production seems to also be
reinforced in the upwelling area. The extent to which those changes can be attributed to
natural decadal variability in this EBUS over interannual timescales remains unclear and
more years of monthly data should be recorded. Sustained volunteer observing lines are
shown as one of the most significative contributors to the knowledge of how ocean
surface waters are being affected by present and future climate change. The results from
VOS lines can provide accurate data for changes in SST, $FCO_2$ and, consequently,
upwelling intensification effects due to global change conditions under decadal natural
variability.





**Data availability.**
All data are free available at the SOCAT data base, http://www.socat.info/ and at the
Carboocean and Carbochange web pages www.CarboOcean.org,
https://carbochange.b.uib.no/, respectively

**Team List**
Melchor Gonzalez Davila, Professor of Marine Chemistry at the University of Las Palmas
de Gran Canaria. Melchor.gonzalez@ulpgc.es
J. Magdalena Santana Casiano, Professor of Chemical Oceanography at the University of
Las Palmas de Gran Canaria. Magdalena.santana@ulpgc.es
Francisco Machin, Associated Professor of Physical Oceanography at the University of
Las Palmas de Gran Canaria. Francisco.machin@ulpgc.es

**Author contributions**
M.G.D. and J.M.S.C worked in the equipment installation, data collection and designed
the study. F.M. processed the data, generated figures and results. All of them collaborated
in the discussion of the data and the writing of the paper.

**Competing interests**
There is not any competing interest.
**Acknowledgements**
Financial support from the European Union through the Integrated Project FP6
CarboOcean under grant agreement no. 511106-2 and FP7 project CARBOCHANGE
under grant agreement no. 264879 are gratefully acknowledged. Special thanks go to the



Mediterranean Shipping Company (MSC) (years 2005- 2008) and the Maersk Company
(years 2010-2013) who provided the ship platforms and scientific facilities. The Modis-
Aqua Ocean Color Data, 2005-2012 reprocessing, NASA OB.DAAC, Greenbelt, MD,
USA is strongly acknowledged.



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
