# Peer review of "Changes in the partial pressure of carbon dioxide in the Mauritanian-Cape Verde"

_Biogeosciences, 2017_

## Referee Comment (RC1) · Anonymous Referee #1 · 6 Apr 2017

Overall evaluation: The authors present many years of Carbon-VOS data from an important upwelling region of the Atlantic. They consider changes in the upwelling index in the surface waters along the route and consequent changes in carbon uptake (and changes in pH). They make some important observations of a decrease in SST and trends in seawater fCO2 that can only be explained by an increased upwelling. They consider both inter-annual variability (and the influence of the NAO) and spatial (latitudinal) variability in the data.

Specific Comments: The figures are clear although I would suggest some minor changes to improve on this. For example in discussion of Figure 2 there is reference to the months (eg: March to September) which cannot really be distinguished in the

figure (that only shows a tick mark for January). This could be improved. Figure 3 is missing the Y-axis label (the X-axis label could also be improved by moving the units into the legend and keeping the x-axis label as 'change per year' for example). The legend should also specify the years shown. In Figure 4 the colours used are not explained in the legend (the shift in season discussed is reliant on knowing what the colours represent). There is a clear typo in the legend suggesting 'pannels' rather than 'panels'. Figure 5 shows normalised fCO2 which I hadn't seen in the method section. The legends in Figure 4 and 5 refer to 'Experimental' series rather than 'in situ' data which may be a person preference but I think 'experimental' is misleading. Figure 7 is in mmol m-3 unit when the text discusses mol m-3.

The text is generally well written and I would suggest that just a few inconsistencies are addressed. For example in the Abstract line 28 refers to an increase in outgassing then in line 30 the authors refer to 'this increase in CO2 intake'. Throughout there is reference to 'opportunity ships', which are more easily recognised to the community as Ships of opportunity (or Carbon-VOS). When fCO2 first appears (line 91) it is referred to as partial pressure rather than fugacity. The oxygen optode is mentioned on line 123 but the location of this sensor is not indicated (at intake or point of CO2 measurements?)

Line 220 suggests a shift from high to strong upwelling, that could be quantified. Line 250 suggests that SACW is nutrient rich, which needs a reference. Claims in line 263 re: the position of the ITCZ in winter also require a reference. Line 449 refers to the EA (in the conclusions but not discussed elsewhere). Anomaly fields for temperature and salinity are mentioned and discussed (line 291) but the data are not shown, perhaps they should be shown/referred to in the supplementary section if they warrant discussion?

Technical corrections: Line 73 correct to wind databases Line 111 Correct to: On its northbound route the ship follows the same track in reverse. Line 125: change among ships to between ships. Line 224: change to annual basis followed that shown by...

Line 272: change arriving to arrival Line 358: Are the authors referring to annual bias or annual based (as stated)? Also change 'upwell' to 'upwelling'. Line 395: change the sentence to read that the pH rate 'is determined'.

---

## Referee Comment (RC2) · Anonymous Referee #2 · 29 May 2017

Review The authors presen a 7 year data record of surface ocean fCO2 observations in the upwelling region off Northwest Africa. The manuscript is a solid piece of science and the data are well described. It is a very nice example how valuable data from carbon-VOS lines can be and how much we can learn from sutainable observations. The work fits perfectly in the scope of "Biogeosciences". This said I have one major criticism: The "Results and Discussion" part is very detailed and very long, which is not bad itself. In contrast, the conclusions are very short and the main point of the paper gets a little bit diluted. The paper reads a little bit like a data paper with a lot of description and only a small part of new knowledge. One idea would be having a short summary of the major findings in each part of section 3. Then taking this summaries

and clearly highlight the major findings of this manuscript. Furthermore I suggest that the authors ask a native English speaker for proof reading.

Specific comments:

line 42: the authors call it here "opportunity ship" but later VOS line. I think VOS line is a commonly used term, so I suggest to indroduce it here and the n use it consequently in the rest of the paper.

Line 58: . . . researcheRs . . .

Line 67: use "wind speed" instead of only "wind"

Line 68: . . . supportS . . .

Line 73/74: . . . different wind databases for the . . .

Line 88: . . .QUIMA-VOS line . . .

Line 102: The term "VOS" is not yet introduced

Line 118: call is VOS line

Line 125: us ". . . seawater flow but varied . . ." instead of ". . . seawater flux used but varied . . ."

Line 133: ". . . imposed . . ."

Lines 136-140: The authors report an accuracy of 1 $\mu$atm but the standard gases don't bracket the expected fCO2 values. This is not according to SOP's and at least for the values outside the standard range the accuracy estimate should be lower.

Lines 154 ff.: To my knowledge the parameterization of Wanninkhoff (1992) has some problems and shouldn't be used. I suggest using Nightingale 2000 which has shown to produce robust flux estimates.

Line 161: what does the * mean?

Lines 166 ff: Is it proved that the general seasonal pattern follows such an sinodial approach?

Lines 222 ff.: Sentence sounds odd. Please rephrase.

Line 225: The authors mention a confidence interval of 9m2 s-1 for the upwelling index (UI) but the scale of in Figure 3 a) is on the order of 10e-3 m2 s-1. This looks like a quite large confidence interval to me.

Line 240 ff.: "South of 15°N . . ." ; This conclusion is not clear to me. Can you please explain why a decreasing UI comes directly from an increasing SST!?

Line 265: use SST and SSS

Lines 307/308: Sounds odd, please rephrase

Line 312: "seasons" instead of "sea1sons"

Lines 324 ff.: 4.3% would translate into an error between 15 and 26 $\mu$atm. I think its better to give the range since the dataset includes data over a wide range.

Lines 330 ff.: The word "and" in line 330 can be deleted. I don't really understand the meaning of comparing NfCO2 (which is temp independent) with temperature again. Please explain the deeper meaning.

Line 368: "average" instead of "averate"

Line 458 ff.: sounds odd, please rephrase

Line 466-469: I don't understand the sentence. Please rephrase.

Line 472 ff.: The authors mention the dependence of the upwelling region on climate change forcing. Please say here clearly which forcing you mean and what the exact effect is.

Line 484: Use "VOS line"

Figure 2:

- please say if red or blue denotes upwelling events. It is somehow clear but this information would improve the readability.

Figure 3:

- are the units correct (see also comment above regarding the confidence interval)

- please add Latidude in °N to the y-axix

- line 228 and 229: "panel" instead of "pannel"

- the authors use "year-1" and "/year" in the same figure. Please use it consistent.

Figure 4:

- Can you name the month that you used for summer, fall, winter and spring?

Figure 5:

- Instead of showing fCO2 (measured) and NfCO2 I would suggest just showing the difference (NfCO2 – FCO2 (Measured)). Then there is no need to compare panel by panel, instead the deviations would be clearly visible.

Figure 7:

- Can you name the month that you used for summer, fall, winter and spring?

---

## Author Comment (AC1) · 8 Jun 2017

Overall evaluation: The authors present many years of Carbon-VOS data from an important upwelling region of the Atlantic. They consider changes in the upwelling index in the surface waters along the route and consequent changes in carbon uptake (and changes in pH). They make some important observations of a decrease in SST and trends in seawater fCO2 that can only be explained by an increased upwelling. They consider both inter-annual variability (and the influence of the NAO) and spatial (latitu- dinal) variability in the data.

Specific C omments: T he fi gures ar e cl ear al though I wo uld su ggest so me minor changes to improve on this. For example in discussion of Figure 2 there is reference to the months (eg: March to September) which cannot really be distinguished in the

figure (that only shows a tick mark for January). This could be improved. Figure 3 is missing the Y-axis label (the X-axis label could also be improved by moving the units    into the legend and keeping the x-axis label as 'change per year' for example). The  legend should also specify the years shown. In Figure 4 the colours used are not explained in the legend (the shift in season discussed is reliant on knowing what the colours represent). There is a clear typo in the legend suggesting 'pannels' rather than 'panels'. Figure 5 shows normalised $fCO_2$ which I hadn't seen in the method section.   The legends in Figure 4 and 5 refer to 'Experimental' series rather than 'in situ' data   which may be a person preference but I think 'experimental' is misleading.  Figure 7 is    in mmol m-3 unit when the text discusses mol m-3.

The figures will be changed according with the reviewer's comments. Typos, explanation about normalised data and other definitions will be considered.

The text is generally well written and I would suggest that just a few inconsistencies      are addressed. For example in the Abstract line 28 refers to an increase in outgassing then in line 30 the authors refer to 'this increase in $CO_2$ intake'.  Throughout there is reference to 'opportunity ships', which are more easily recognised to the commu- nity as Ships of opportunity (or Carbon-VOS). When $fCO_2$ first appears (line 91) it is referred to as partial pressure rather than fugacity. The oxygen optode is mentioned on line 123 but the location of this sensor is not indicated (at intake or point of $CO_2$ measurements?)

We will change the terms to account for the real trends, VOS will be considered along the text as indicated, $fCO_2$ will be defined properly and the position of the optode (after the seawater pump, the intake is divided in two lines, one feeding the $pCO_2$ system and the other the oxygen sensor, the fluorometer and the seabird thermosalinometer).

Line 220 suggests a shift from high to strong upwelling, that could be quantified.

The values for the upwelling indexes will be included in order to show this change, following the data on Figure    2.

Line 250 suggests that SACW is nutrient rich, which needs a reference.

The reference will be added.

[Figure]

Claims in line 263 re: the position of the ITCZ in winter also require a reference
Line 272: change arriving to arrival Line 358: Are the authors referring to annual bias or annual based (as stated)? Also change 'upwell' to 'upwelling'. Line 395: change the sentence to read that the pH rate 'is determined'.

All these technical aspects will be changed as indicated.

[Figure]

[Figure]
Review The authors presen a 7 year data record of surface ocean fCO2 observations in the upwelling region off Northwest Africa. The manuscript is a solid piece of science and the data are well described. It is a very nice example how valuable data from carbon-VOS lines can be and how much we can learn from sutainable observations. The work fits perfectly in the scope of "Biogeosciences". This said I have one major criticism: The "Results and Discussion" part is very detailed and very long, which is not bad itself. In contrast, the conclusions are very short and the main point of the paper gets a little bit diluted. The paper reads a little bit like a data paper with a lot of description and only a small part of new knowledge. One idea would be having a short summary of the major findings in each part of section 3. Then taking this summaries

and clearly highlight the major findings of this manuscript. Furthermore I suggest that the authors ask a native English speaker for proof reading.

We will follow the idea indicated by the reviewer including a short summary of the major finding in each part of section 3. The text will be reviewed by a native English speaker.

Specific comments:

line 42: the authors call it here "opportunity ship" but later VOS line. I think VOS line is a commonly used term, so I suggest to indroduce it here and the n use it consequently in the rest of the paper.

This will be done

Line 58: . . . researcheRs . . .

It will be changed

Line 67: use "wind speed" instead of only "wind"

Done

Line 68: . . . supportS . . .

*Done*

Line 73/74: . . . different wind databases for the . . .
*done*

Line 88: . . .QUIMA-VOS line . .

*done.*

Line 102: The term "VOS" is not yet introduced

It will be introduced

Line 118: call is VOS line

done

[Figure]

Line 125: us ". . . seawater flow but varied . . ." instead of ". . . seawater flux used but varied . . ."

done

Line 133: ". . . imposed . . ."

done

Lines 136-140: The authors report an accuracy of 1 $\mu$atm but the standard gases don't bracket the expected $fCO_2$ values. This is not according to SOP's and at least for the values outside the standard range the accuracy estimate should be lower.

This will be improved and considered

Lines 154 ff.: To my knowledge the parameterization of Wanninkhoff (1992) has some problems and shouldn't be used. I suggest using Nightingale 2000 which has shown to produce robust flux estimates.

We agree with reviewer´s comments. We have used W92 because most of the computed fluxes used that parameterization. We will present the data with Nightingale parameterization.

Line 161: what does the * mean?

We will explain that (the fitted values)

Lines 166 ff: Is it proved that the general seasonal pattern follows such an sinodial approach?

We have included the errors for that fitting procedure for all variables. We have tried other functions including even the SST but there was not a significant increase in the estimation. We will also present the fitting between real and estimated value in order to provide the goodness of the fitting. Moreover, the data also shows how the values moves with seasons along the years. We used this model also followed by others, as Lüger et al., 2004 and in time series of data.

Lines 222 ff.: Sentence sounds odd. Please rephrase.

This will be rewritten.

Line 225: The authors mention a confidence interval of 9m2 s-1 for the upwelling index (UI) but the scale of in Figure 3 a) is on the order of 10e-3 m2 s-1. This looks like a quite large confidence interval to me.

It looks like we did not write in a clear way the values. The UI are in the order of hundreds of $m^2s^{-1}$ and we indicated that UI*$10^{-3}$ it is what is plotted and you have values of 0.1 to 0.3, that is until 300. A confidence interval of 9 is expected. We will write this in a more readable way.

Line 240 ff.: "South of 15˚N . . ." ; This conclusion is not clear to me. Can you please explain why a decreasing UI comes directly from an increasing SST!?

A detail explanation is included. UI were computed without temperature data. As UI decreases, less cool waters arrive to the surface: This fact together with the advection of warm water along the coast produce an increasing SST. The word "indicate" has been changed.

Line 265: use SST and SSS

Done

Lines 307/308: Sounds odd, please rephrase

This will be rewritten

[Figure]

Line 312: "seasons" instead of "sea1sons"

Done

Lines 324 ff.: 4.3% would translate into an error between 15 and 26 $\mu$atm. I think its better to give the range since the dataset includes data over a wide range.
Done

Lines 330 ff.: The word "and" in line 330 can be deleted. I don't really understand the meaning of comparing NfCO2 (which is temp independent) with temperature again. Please explain the deeper meaning.

Done . The sentence "NfCO$_2^{sw}$ was related with SST in order to account for effects do not removed after normalization". This was included

Line 368: "average" instead of "averate"

Done

Line 458 ff.: sounds odd, please rephrase

Done

Line 466-469: I don't understand the sentence. Please rephrase.

We have changed the sentence.

Line 472 ff.: The authors mention the dependence of the upwelling region on climate change forcing. Please say here clearly which forcing you mean and what the exact effect is.

We will include references and some aspects indicated in the introduction.

Line 484: Use "VOS line"

Done

Figure 2:

- please say if red or blue denotes upwelling events. It is somehow clear but this information would improve the readability.

- Done

Figure 3:

- are the units correct (see also comment above regarding the confidence interval)

- please add Latidude in °N to the y-axix

- line 228 and 229: "panel" instead of "pannel"

- the authors use "year-1" and "/year" in the same figure. Please use it consistent.

Figure 3 will be improved including those aspects (also that indicated by reviewer 1)

- Figure 4:

- Can you name the month that you used for summer, fall, winter and spring?

Done

- Figure 5:

- Instead of showing fCO2 (measured) and NfCO2 I would suggest just showing the difference (NfCO2 – FCO2 (Measured)). Then there is no need to compare panel by panel, instead the deviations would be clearly visible.

We will keep fCO2 (measured) because it will be important to improve the readability. We will include the difference as indicated to see deviations.

Figure 7:

- Can you name the month that you used for summer, fall, winter and spring?
  Done

---

## Author Response (AR1)

Overall evaluation: The authors present many years of Carbon-VOS data from an important upwelling region of the Atlantic. They consider changes in the upwelling index in the surface waters along the route and consequent changes in carbon uptake (and changes in pH). They make some important observations of a decrease in SST and trends in seawater fCO2 that can only be explained by an increased upwelling. They consider both inter-annual variability (and the influence of the NAO) and spatial (latitu- dinal) variability in the data.

Specific C omments: T he fi gures ar e cl ear al though I wo uld su ggest so me minor changes to improve on this. For example in discussion of Figure 2 there is reference to the months (eg: March to September) which cannot really be distinguished in the

figure (that only shows a tick mark for January). This could be improved. Figure 3 is missing the Y-axis label (the X-axis label could also be improved by moving the units into the legend and keeping the x-axis label as 'change per year' for example). The legend should also specify the years shown. In Figure 4 the colours used are not explained in the legend (the shift in season discussed is reliant on knowing what the colours represent). There is a clear typo in the legend suggesting 'pannels' rather than 'panels'. Figure 5 shows normalised $fCO_2$ which I hadn't seen in the method section. The legends in Figure 4 and 5 refer to 'Experimental' series rather than 'in situ' data which may be a person preference but I think 'experimental' is misleading. Figure 7 is in mmol m-3 unit when the text discusses mol m-3.

The figures were changed according with the reviewer's comments. Typos, explanation about normalised data and other definitions were considered.

The text is generally well written and I would suggest that just a few inconsistencies are addressed. For example in the Abstract line 28 refers to an increase in outgassing then in line 30 the authors refer to 'this increase in $CO_2$ intake'. Throughout there is reference to 'opportunity ships', which are more easily recognised to the community as Ships of opportunity (or Carbon-VOS). When $fCO_2$ first appears (line 91) it is referred to as partial pressure rather than fugacity. The oxygen optode is mentioned on line 123 but the location of this sensor is not indicated (at intake or point of $CO_2$ measurements?)

We have changed the terms to account for the real trends, VOS was considered along the text as indicated, $fCO_2$ was defined properly and the position of the optode indicated(after the seawater pump, the intake is divided in two lines, one feeding the $pCO_2$ system and the other the oxygen sensor, the fluorometer and the seabird thermosalinometer).

Line 220 suggests a shift from high to strong upwelling, that could be quantified.

The values for the upwelling indexes were included in order to show this change, following the data on Figure 2.

Line 250 suggests that SACW is nutrient rich, which needs a reference.

The reference was added.

[Figure]

Claims in line 263 re: the position of the ITCZ in winter also require a reference Line 272: change arriving to arrival Line 358: Are the authors referring to annual bias or annual based (as stated)? Also change 'upwell' to 'upwelling'. Line 395: change the sentence to read that the pH rate 'is determined'.

All these technical aspects were changed as indicated.

[Figure]

Biogeosciences Discuss.,
doi:10.5194/bg-2017-83-RC2, 2017

[Figure]

Review The authors presen a 7 year data record of surface ocean fCO2 observations in the upwelling region off Northwest Africa. The manuscript is a solid piece of science and the data are well described. It is a very nice example how valuable data from carbon-VOS lines can be and how much we can learn from sutainable observations. The work fits perfectly in the scope of "Biogeosciences". This said I have one major criticism: The "Results and Discussion" part is very detailed and very long, which is not bad itself. In contrast, the conclusions are very short and the main point of the paper gets a little bit diluted. The paper reads a little bit like a data paper with a lot of description and only a small part of new knowledge. One idea would be having a short summary of the major findings in each part of section 3. Then taking this summaries

and clearly highlight the major findings of this manuscript. Furthermore I suggest that the authors ask a native English speaker for proof reading.

We have followed the idea indicated by the reviewer including a short summary of the major finding in each part of section 3. The text was reviewed by a native English speaker.

Specific comments:

line 42: the authors call it here "opportunity ship" but later VOS line. I think VOS line is a commonly used term, so I suggest to indroduce it here and the n use it consequently in the rest of the paper.

Done

Line 58: . . . researcheRs . . .

Changed

Line 67: use "wind speed" instead of only "wind"

Done

Line 68: . . . supportS . . .

*Done*

Line 73/74: . . . different wind databases for the *. . .*
*done*

Line 88: . . .QUIMA-VOS line *. .*

*done.*

Line 102: The term "VOS" is not yet introduced

This was introduced

Line 118: call is VOS line done

[Figure]

Line 125: us "*. . . seawater flow but varied . . .*" instead of "*. . . seawater flux used but varied . . .*"

done

Line 133: "*. . . imposed . . .*"

done

Lines 136-140: The authors report an accuracy of 1 $\mu$atm but the standard gases don't bracket the expected fCO2 values. This is not according to SOP's and at least for the values outside the standard range the accuracy estimate should be lower.

This was improved and considered

Lines 154 ff.: To my knowledge the parameterization of Wanninkhoff (1992) has some problems and shouldn't be used. I suggest using Nightingale 2000 which has shown to produce robust flux estimates.

We agree with reviewer´s comments. We have used W92 because most of the computed fluxes used that parameterization. We present the data with Nightingale parameterization.

Line 161: what does the * mean?

This has been explained (the fitted values)

Lines 166 ff: Is it proved that the general seasonal pattern follows such an sinodial approach?

We have included the errors for that fitting procedure for all variables. We have tried other functions including even the SST but there was not a significant increase in the estimation. We present the fitting between real and estimated value in order to provide the goodness of the fitting. Moreover, the data also shows how the values moves with seasons along the years. We used this model also followed by others, as Lüger et al., 2004 and in time series of data.

Lines 222 ff.: Sentence sounds odd. Please rephrase.

This was rewritten.

Line 225: The authors mention a confidence interval of 9m2 s-1 for the upwelling index (UI) but the scale of in Figure 3 a) is on the order of 10e-3 m2 s-1.  This looks like a   quite large confidence interval to me.

It looks like we did not write in a clear way the values. The UI are in the order of hundreds of $m^2s^{-1}$ and we indicated that UI*$10^{-3}$ it is what is plotted and you have values of 0.1 to 0.3, that is until 300. A confidence interval of 9 is expected. We have wrotten this in a more readable way.

Line 240 ff.: "South of 15˚N . . ." ; This conclusion is not clear to me. Can you please explain why a decreasing UI comes directly from an increasing SST!?

A detail explanation is included. UI were computed without temperature data. As UI decreases, less cool waters arrive to the surface: This fact together with the advection of warm water along the coast produce an increasing SST. The word "indicate" has been changed. The paragraph have been improved.

Line 265: use SST and SSS

Done

Lines 307/308: Sounds odd, please rephrase

This was rewritten

[Figure]

Line 312: "seasons" instead of "sea1sons"

Done

Lines 324 ff.: 4.3% would translate into an error between 15 and 26 $\mu$atm. I think its better to give the range since the dataset includes data over a wide range.
Done

Lines 330 ff.: The word "and" in line 330 can be deleted. I don't really understand the meaning of comparing NfCO2 (which is temp independent) with temperature again. Please explain the deeper meaning.

Done . The sentence "$NfCO_2^{sw}$ was related with SST in order to account for effects do not removed after normalization". This was included

Line 368: "average" instead of "averate"

Done

Line 458 ff.: sounds odd, please rephrase

Done

Line 466-469: I don't understand the sentence. Please rephrase.

We have changed the sentence.

Line 472 ff.: The authors mention the dependence of the upwelling region on climate change forcing. Please say here clearly which forcing you mean and what the exact effect is.

This was rewritten, and the processes were indicated in the introduction part.

Line 484: Use "VOS line"

Done

Figure 2:

- please say if red or blue denotes upwelling events. It is somehow clear but this information would improve the readability.

- Done

Figure 3:

- are the units correct (see also comment above regarding the confidence interval)

- please add Latidude in °N to the y-axix

- line 228 and 229: "panel" instead of "pannel"

- the authors use "year-1" and "/year" in the same figure. Please use it consistent.

Figure 3 (now 4) was improved including those aspects (also that indicated by reviewer 1)

- Figure 4:

- Can you name the month that you used for summer, fall, winter and spring?

Done

- Figure 5:

- Instead of showing fCO2 (measured) and NfCO2 I would suggest just showing the difference (NfCO2 – FCO2 (Measured)). Then there is no need to compare panel by panel, instead the deviations would be clearly visible.

We have kept fCO2 (measured) because it will be important to improve the readability. We have included the difference as indicated to see deviations.

Figure 7:

- Can you name the month that you used for summer, fall, winter and spring?
Done

[revised manuscript text omitted]

Fig. 2

[Figure]

Fig. 34

[Figure]

[Figure]

[Figure]

[Figure]

Fig. 6

[Figure]

Fig. 7

[Figure]

---

## Author Response (AR2)

**Associate Editor Decision: Publish subject to minor revisions (Editor review)** (25 Jul 2017) by Kai G. Schulz
Comments to the Author:
Dear authors, congratulations, your revisions have been approved by the two referees of your original submission and hence I will accept your paper, pending some minor amendments.

1) Could you please specify whether the the plots in figure 5 are 'measured-normalized' or the other way around.

2) One referee noted that not all the presented data is available in SOCAT and that only the cruises between 2011 and 2012 were found. Could you please check and add the missing data. In case this data can be found in other databases I suggest adding this information to the table in the supplement.

3) On this note, one referee was wondering why the authors have chosen not to utilize additional data available in SOCAT for this region. May I suggest adding a paragraph discussing this issue and potential implications.

We have specified the difference.

We have indicated the data are public. I have confirmed that in the Carboocean and Carbochange project all data are available and I have asked the responsible (Benjamin Pfeil) to confirm me all of them have been made public in SOCAT database. He is now on holidays, but I have asked him to check that and if not all them are public, please make it.

We agree with the reviewer that more data exist in the region in SOCAT. We have used only the QUIMA data because all of them keep the same line and distance to the coast avoiding physical effects in the partial pressure of $CO_2$ associated with the distance to the upwelling cells. This has been indicated in the paper in the introduction.

[revised manuscript text omitted]

Fig. 2

[Figure]

Fig. 3

[Figure]

Fig. 4

[Figure]

[Figure]

[Figure]

Fig. 6

[Figure]

Fig. 7

[Figure]